# State-of-the-Art Review on Inhalable Lipid and Polymer Nanocarriers: Design and Development Perspectives

**DOI:** 10.3390/pharmaceutics16030347

**Published:** 2024-03-01

**Authors:** Gabriella Costabile, Gemma Conte, Susy Brusco, Pouria Savadi, Agnese Miro, Fabiana Quaglia, Ivana d’Angelo, Francesca Ungaro

**Affiliations:** 1Laboratory of Drug Delivery, Department of Pharmacy, University of Napoli Federico II, Via Domenico Montesano 49, 80131 Napoli, Italy; gabriella.costabile@unina.it (G.C.); gemma.conte@unicampania.it (G.C.); susy.brusco@unina.it (S.B.); miro@unina.it (A.M.); quaglia@unina.it (F.Q.); ungaro@unina.it (F.U.); 2DiSTABiF, University of Campania “Luigi Vanvitelli”, 81100 Caserta, Italy; pouria.savadisomeeh@unicampania.it

**Keywords:** pulmonary drug delivery, lung barriers, inhaled nanomedicines, polymer nanoparticles, lipid nanoparticles, airway mucus

## Abstract

Nowadays, the interest in research towards the local administration of drugs via the inhalation route is growing as it enables the direct targeting of the lung tissue, at the same time reducing systemic side effects. This is of great significance in the era of nucleic acid therapeutics and personalized medicine for the local treatment of severe lung diseases. However, the success of any inhalation therapy is driven by a delicate interplay of factors, such as the physiochemical profile of the payload, formulation, inhalation device, aerodynamic properties, and interaction with the lung fluids. The development of drug delivery systems tailored to the needs of this administration route is central to its success and to revolutionize the treatment of respiratory diseases. With this review, we aim to provide an up-to-date overview of advances in the development of nanoparticulate carriers for drug delivery to the lung tissue, with special regard concerning lipid and polymer-based nanocarriers (NCs). Starting from the biological barriers that the anatomical structure of the lung imposes, and that need to be overcome, the current strategies to achieve efficient lung delivery and the best support for the success of NCs for inhalation are highlighted.

## 1. Introduction

The pulmonary route is currently regarded as the administration route of choice for the local treatment of severe lung diseases since: (i) it directly acts on the target and (ii) achieves high local drug availability, while limiting systemic toxicity. Meanwhile, lungs may represent the port of entry for the needle-free systemic administration of macromolecules as well as for small molecules, to obtain the rapid onset of action, low metabolism and high bioavailability [1]. Nevertheless, some disadvantages still exist and are mainly related to the need of opportunely engineered inhalable drug particles able to maximize the amount of drug deposited and/or absorbed in the lung, while shielding its unfavorable interactions with the pulmonary environment.

Although carrier-free drug particles raise fewer safety concerns for inhalation, biocompatible materials, such as lipids and biodegradable polymers, may provide the efficient protection and delivery of a variety of therapeutics, spanning from antibiotics, headed for resistant biofilm bacteria, and gene materials, directed inside airway epithelial cells. Thus, a growing interest has been especially devoted to the application of nanoparticulate carriers for inhalation. Recent advances in materials, particle engineering techniques, and inhalation devices enabled the formulation of nanocarriers (NCs) for the pulmonary delivery of drugs with an improved ability to land in the lungs, to overcome lung barriers and to fulfill specific therapeutic needs [2,3]. Indeed, the peculiar properties of NCs, including their small size, confer them with the special ability to surpass the mucus barrier lining lung epithelium and to gain access to the cell target. Based on the constituents more widely used, the carriers can be classified as organic (i.e., polymers, lipids) and inorganic (i.e., metals, quantum dots, calcium phosphate, oxide, and silica) [4]. 

Despite being promising, the design and development of NCs for inhalation is not straightforward. The efficient lung deposition of inhaled drugs is a very complex task, resulting from the interplay of several factors. It depends not only on NCs’ size and morphology, but also on the inhalation device, the breathing pattern, and even the lung anatomy of the individual [5,6]. Acknowledging that efficient deposition in the lungs is an essential prerequisite for successful pulmonary drug delivery, there is also an increasing need to control what happens after the drug particles have landed. After deposition in the lung, the success of any inhalation therapy will strongly depend upon (i) drug permeation through airway mucus; (ii) the drug interaction with the cell target; and (iii) macrophage clearance escape. The drug’s capacity to infiltrate bacterial biofilm is especially crucial in the context of antimicrobials [7].

The aim of this review is to provide an overview on recent advances in developing NCs for drug inhalation. After a brief description of the main factors governing the deposition of drugs in the lungs and their fate after landing, the advantages of engineered NCs for inhalation are described. Then, an update on inhalable biocompatible lipids and polymeric and hybrid NCs is presented. Despite their intriguing properties, some of the most studied inorganic carriers for pulmonary applications have been associated with the largest records of pulmonary toxicity. For this reason, they will be not the object of this review [8]. An exhaustive review on the topic highlighting the potential for the future prospective of inorganic materials for inhalation are reported [8,9].

## 2. Factors Governing Drug Deposition in the Lung

A better understanding of the mechanisms and factors determining aerosol deposition is of upmost importance to provide for the maximum and reproducible amounts of drug deposited in the desired region of the lungs. As well known, three primary mechanisms account for drug deposition along the respiratory tract, namely inertial impaction, gravitational sedimentation, and Brownian diffusion [10]. They are complemented by electrostatic attractions and particle interception (in case of elongated particles) and are likely affected by the air flow turbulence. 

The particle mass median aerodynamic diameter (MMAD), resulting from the size, density, and shape of the particle, critically influences the mechanism and the site of drug deposition [11,12]. In detail, particles with an MMAD larger than 10 µm are mainly deposited by inertial impaction in the extra-thoracic region and in the bifurcations. Indeed, particles become more and more inert with their growing size and their ability to follow the respiratory flow is reduced proportionally to the flow rate. If pharmaceutical aerosols penetrate the small conducting airways, the dominant deposition mechanism is gravitational sedimentation (i.e., particles fall under gravity onto the airway walls). Both impaction and sedimentation cause the deposition of particles larger than 3–5 µm in the smaller airways (i.e., bronchus, bronchioles) before reaching the alveoli. Differently, particles with an MMAD in the 1–2 µm range will likely deposit into the capillary-rich alveolar airspaces, which represent the target for systemic drug delivery through the lungs. Here, deposition is mainly influenced by Brownian diffusion, which dominates for particles with diameters of less than 1 µm. Particles with an MMAD between 0.1 and 1 μm, such as NCs, are mostly exhaled, even though ultrafine particles (lower than 100 nm) may paradoxically deposit in the respiratory tract taking advantage of their random Brownian motion.

Inhalation devices significantly influence the pattern of drug deposition within the lungs [10]. The increasing research interest towards inhaled therapeutics has recently led to tremendous innovations in designing inhalation devices able to ensure a high aerosolization performance, consistent therapeutic efficacy, and satisfactory patient adherence to treatment [13,14]. Vibrating mesh and software technologies have resulted in nebulizers having highly accurate dosing and portability. On the other hand, advanced particle engineering techniques harnessed dry powder inhalers (DPIs) for also delivering high-dose drugs, such as antibiotics. Though new propellant systems attempt to improve the performance of aerosols delivered by pressurised metered dose inhalers (pMDIs), drug-propellant incompatibility and delivering high-dose drugs are still important technical constraints. Detailed reviews have recently summarized all the advantages and challenges of the different inhalation devices [13,15].

Inhalation flow and the velocity at which aerosol particles are emitted from the device and travel through the airways also strongly impact pulmonary deposition patterns [15,16]. As a rule, faster inhalation increases the inertial impaction of aerosols in the oropharynx and in bifurcations in the large central airways, whereas slower inhalation results in more peripheral deposition patterns. Nevertheless, when using a low-resistance passive DPI, slow inhalation may be insufficient to disaggregate the inhaled powder particles, and can therefore limit lung deposition. Thus, lung deposition from DPIs paradoxically increases as inspiratory flow increases. Overall, an optimal design strategy for inhaled drugs should consider both the inhalation device and the inhalation flow rate [16,17].

## 3. Overcoming Lung Barriers through Tailored Nanocarriers

Predicting the fate of inhaled drugs after landing in the lung is key to formulation development, but it still represents a relatively complex issue. An in-depth knowledge of the barriers imposed by the route of administration appears mandatory to improve the lung availability of drugs and, thus, the therapeutic outcome. The lung barriers can be grouped into two main categories: the non-cellular and cellular barriers (Figure 1) [18,19,20]. 

Lung-lining fluids, such as airway mucus and lung surfactant, represent the main non-cellular barriers. They strongly affect the behavior of drug particles in the lung determining their solubility, diffusion, permeation, and, in so doing, drug bioavailability. Underneath the lung-lining fluids, the human airway epithelium represents the main cell barrier to drug transport towards its intracellular target and systemic absorption. Cell barriers also comprise the immune system’s cells, such as macrophages, which can uptake inhaled drug particles, reducing their availability and effectiveness in the lung [21].

For some respiratory diseases, other barriers need to be considered, such as the bacterial biofilm in lung infections. In this case, the effectiveness of inhaled antimicrobials can be dramatically reduced by the interactions with the component of the biofilm, such as polysaccharides and proteins, leading to the drug resistance mechanism (i.e., antibiotic resistance) [22,23]. 

The success of NCs for drug inhalation depends upon the possibility of tailoring the composition, size, surface charge, and morphology of the particle to overcome the barrier imposed by the lung and to gain access to the specific target [15].

### 3.1. Non-Cellular Barriers

The human lung epithelium is covered by the lung-lining fluid (LLF), a primary and heterogeneous constituent of the pulmonary host defense system. The two essential elements of the LLF are the airway surface liquid (ASL), a mucus gel–aqueous sol complex lining conductive airways, and the alveolar subphase fluid (AVSF), at the alveolar level.

Airway mucus is one of the most investigated non-cellular barriers affecting in situ permanence and the extent of absorption of the inhaled drug particles [24]. Highly cross-linked mucin chains create a dense porous structure, with the thickness and porosity being variable for the lung area and pathological conditions. Two major mechanisms may stop particles from readily diffusing through mucus gel, which are “size filtering” through the mucus meshes and “interaction filtering” [24]. Insoluble particles might be filtered by the mucus gel layer if bigger than the mesh-spacing of the mucin network, or establish hydrophobic, electrostatic, and/or hydrogen bonding interactions with the negatively charged mucin chains. Trapped particles are moved toward the pharynx, and ultimately to the gastrointestinal tract, by the upward movement of mucus generated by the cilia beating (i.e., mucociliary clearance) [25].

NCs’ properties, including their size, charge, and hydrophobicity, are crucial in determining drug diffusion through mucus [26,27]. The pore size of the mucus gel layer ranges from 100 to 200 nm, suggesting that only NCs in this size range could potentially penetrate. Nevertheless, significant healthy-to-diseased and/or patient-to-patient variations are reported. It is nowadays acknowledged that a sufficiently hydrophilic and uncharged surface may minimize the adhesive interactions between mucin and the NC, thus improving their mucus-penetrating ability [26,28].

The pulmonary fluid layer reduces in thickness throughout the airways, forming single droplets on top of the limited ciliated cells of the lower bronchi and an extremely thin layer of surfactant in the alveoli. The pulmonary surfactant layer prevents alveolar collapse during expiration and is composed of approximately 90% lipids, mainly dipalmitoylphosphatidylcholine (DPPC), and 10% proteins (i.e., surfactant protein A, B, C, and D) [29]. Upon contact with the surfactant, larger sized particles are displaced from the airspace to the hypo-phase due to wetting forces, a phenomenon that probably also occurs with nano-sized particles [30]. In the hypo-phase, the particles may interact with surfactant proteins or may be taken up by alveolar macrophages [30,31]. Many of the literature findings suggest that, depending on their composition and size, inhaled particles may also interfere with the function of the pulmonary surfactant, thus hindering its physiological and essential role in the lung [23,24]. This issue should be properly considered when designing novel inhaled nanomedicines [32,33].

### 3.2. Lung Epithelial Barrier

The interactions of inhaled particles with the human airway epithelial barrier also play a crucial role in determining the availability of drugs in the lung. The lungs are made by many different types of cells, of which the main type are epithelial cells. The thickness and the properties of the lung’s epithelial cell layer vary from a columnar cell monolayer with a thickness of 60 µm in the up airways (i.e., bronchi) to a broad cell monolayer 0.2 µm thick in the alveoli. The alveolar epithelial layer separates the lumen airspace from the pulmonary aqueous interstitial compartment, which is composed by different elements, such as the lymphatic vessel, collagen, and fiber [5,34,35]. 

The uptake of the inhaled drug by the lung’s epithelial cells is influenced by the lung physiopathology [34]. As a matter of fact, while lipophilic drugs are thought to be rapidly absorbed by passive transcellular diffusion through epithelial cells, small hydrophilic compounds likely diffuse across the epithelium through aqueous pores in intercellular gap junctions [36]. Tight junctions, efflux proteins, and cellular enzymes play an important role as barriers in the absorption process as well [36,37]. Of note, tight junctions, mainly located on the apical side of the cell layer, are influenced by pathological conditions [36]. Indeed, the epithelial barrier function may decrease due to the disruption of tight junctions in asthma and chronic obstructive pulmonary diseases (COPD) [38,39], while their function may be enhanced in cystic fibrosis (CF) airway epithelial cells [40,41].

To improve drug accumulation at the cell level, different formulation approaches can be pursued. Drug interactions with cells can be regulated by simply using carriers with adequate size and surface properties, such as charge, hydrophilicity, and a shielding cloud [42,43]. However, efficient drug delivery to the final intracellular target is still challenging [44,45]. This is the case of emerging nucleic acid-based therapeutics (DNA, siRNA, and oligonucleotides), which demand for adequately engineered nanoparticulate systems, comprising not only of biodegradable polymers able to compact, to protect, and to release the entrapped nucleic acid, but also a biomimetic shell containing agents able to facilitate its endo-lysosomal escape (e.g., fusogenic peptides or lipids, endosome destabilizing polymers) [45,46]. The decoration of the carrier surface with different functional motifs can also be attempted to build up actively targeted constructs [47,48,49]. 

In the light of these findings, advanced in vitro models are paramount to design and to develop effective inhaled drugs. During recent years, many cell culture models of nasal, bronchial, and alveolar barriers have been developed, varying from 2D monolayer cultures to advanced 3D co-cultures, with the aim to better resemble what happens in vivo and to provide further insight into cellular responses or interactions with inhaled drug particles [50,51,52,53,54]. The direct aerosolization of the drug formulation on the cells through appropriately designed exposure systems, such as the VITROCELL^®^ Cloud or the PreciseInhale^®^, is feasible to further increase the significance of the biological results [50,55]. Finally, lung-on-chips are emerging to emulate both the morphological features and biological functionality of the airway barrier with the ability to integrate respiratory motions and ensuing tissue strains [54,56,57].

### 3.3. Macrophage-Mediated Clearance

Macrophages are a type of lung-resident immune cells, designed to eliminate any foreign material that reaches the pulmonary environment [58,59,60]. They represent a unique cell population in the peripheral lungs that promptly responds to any airborne irritant or microbe [58]. Drug interactions with macrophages and the subsequent uptake can be enhanced or reduced by adequately tuning the properties of inhaled drug particles [61].

The size of particulate carrier systems significantly influences particle–macrophage interactions, as widely discussed in the literature [62,63]. Particles with a 1–5 µm size are taken up by alveolar macrophages, both in vitro and in vivo, to a greater extent compared to particles that are smaller or larger [64]. Also, the mechanism of particle uptake changes as a function of the carrier size. Micron-sized inhaled particles (1–5 µm) are taken up by alveolar macrophages mainly via active phagocytosis, whereas this is unlikely for nanosized particle [63]. Depending on their size, NCs enter alveolar macrophages by pathways other than phagocytosis: while NCs bigger than 0.2 µm are probably internalized via pinocytosis, smaller carriers (less than 150 nm) can be internalized via calveolae (50–100 nm) or clathrin-mediated (100–120 nm) uptake [63,65]. 

There is clear evidence also of the importance of the shape in governing particle uptake by alveolar macrophages [63]. When particles are internalized, the macrophage membrane spreads around the engulfed particle, and the progression of internalization is dependent on the contact angle between the particle and the membrane [61]. It was successfully demonstrated that rod-shaped nanostructured particles can be internalized with high efficiency (90% of uptake in 48 h) without a cytotoxic effect [66]. In addition, for aspherical particles, the orientation of the particle is crucial. The local shape of the particle at the point of cell attachment, rather than the overall shape, determines cell internalization. For example, if an ellipse-shaped particle is used, a macrophage attached to the pointed end of the ellipse internalizes the NCs in less time than the same attached to a flat region. Spherical particles, in contrast, are symmetrical and thus can be internalized from any point of attachment [67,68]. Similarly, shape-switching elongated particles were quickly engulfed by macrophages once the particles became spherical in shape [69].

Besides particle size and shape, stiffness is emerging as a design parameter for modulating the interaction of NCs with phagocytic cells [70,71]. As a rule, the increased mechanical robustness and overall stiffness of particles leads to increased phagocytosis. Inspired by blood cell behavior, deformable discoidal polymeric nano-constructs have been especially designed to minimize sequestration by phagocytic cells [70,72]. Tailoring nano-constructs’ softness has been demonstrated to be crucial for modulating phagocytic cell sequestration [73]. By three different shapes (circular, elliptical, and quadrangular), two characteristic sizes, and a Young’s modulus varying over two orders of magnitude (from 100 kPa to 10 MPa), professional phagocytic cells were observed to engulf more avidly rigid nano-constructs as compared to soft ones [73].

### 3.4. Bacterial Biofilm 

In the case of lung infections, the effectiveness of inhaled antimicrobials can be seriously limited by the biofilm-producing capacity of some bacteria. Bacterial biofilm is a community of bacteria that is embedded in a self-produced matrix, the extracellular polymeric substance (EPS), composed of polysaccharides, secreted proteins, and extracellular DNAs [74]. Biofilm formation boosts antimicrobial resistance through different mechanisms, such as the sequestration and limited drug diffusion and the increase in antimicrobial efflux pump expression [75]. In particular, the EPS may establish electrostatic/hydrophobic interactions with the antimicrobial drugs, thus limiting drug diffusion towards bacteria and consequent antimicrobial activity [75].

Drug delivery through engineered NCs has been demonstrated to be a promising approach to assist drug diffusion across biofilm towards the bacterial target. Again, the achievement of this goal strongly depends on particle size and surface properties [19]. The results achieved on different biofilm bacteria highlighted how the size of the EPS meshes can make the difference, with a size cut-off for the optimal penetration of polymeric NCs into biofilm clusters, independently of bacteria, of around 100–130 nm [76]. Reduced but effective diffusion across a *P. aeruginosa* biofilm was found in vitro for liposomes with a diameter of 200 and 300 nm, while no diffusion was revealed for liposomes with size of 1 µm [77].

Although particle size plays a role in governing particle diffusion through the biofilm, the complex composition of the EPS matrix results also in several electrostatic/hydrophobic interactions between the NC and biofilm. Thus, the engineering of the NC surface represents another crucial step in the design of the inhaled particulate system. Again, surface charge and hydrophilicity play a pivotal role. As a general rule, while negatively charged particles may interact with positively charged polysaccharides, imparting a positive charge may improve the particle’s interaction with the negative components of the matrix, such as alginates, proteins, and DNA [76]. In fact, the mobility of NCs into the biofilm appears increased for particles with a neutral surface, as in the case of particles with polyethylene glycol (PEG) surface modification (PEGylation) [76,78]. Nevertheless, conflicting data are reported in the literature. Positively charged quantum dots are able to penetrate and diffuse across a bacterial biofilm faster and more efficiently than negative and neutral ones [79,80]. Furthermore, some examples of cationic particles that were efficiently diffused and distributed into the bacterial biofilm are reported in the literature [79,80]. In some cases, positively-charged NCs showed enhanced distribution in the bacterial biofilm and consequent improved antimicrobial activity when compared to negatively charged NCs [81]. Notably, the surface charge may result also in a different location inside the biofilm matrix after diffusion. Actually, charged NCs can be localized close to the bacteria membrane [76]. The differential localization can be attributed to the hydrophilic nature of the NC surface, and more specifically, the higher hydrophobicity of the particle surface may enhance NC colocalization with bacterial cells within the biofilm [82]. Based on this principle, different attempts were pursued to provide an NC surface switch to adapt the NC properties to the target (i.e., mucus before and bacterial biofilm after). This is the case of environment-adaptive NCs developed using d-α-tocopheryl polyethylene glycol 1000 succinate (TPGS) [83,84]. After PEG-assisted mucus penetration, the enzymatic cleavage of PEG chains generates a lipophilic surface that allows the anchoring of the NC to the biofilm, where it serves as a depot for the prolonged exposure of bacteria to antibiotics [83]. 

## 4. Lipid-Based Nanocarriers for Lung Administration: State-of-the-Art

In the field of nanomedicine, lipid-based delivery systems are certainly the most investigated delivery platforms and arguably the most successful one [85,86,87]. Their growth overtime has been slow but steady and punctuated by several important milestones. The first big success has been the approval of Doxil back in 1995. Then, in 2018, the approval of Onpratto by the US FDA as a non-viral gene therapy approach became a watershed in the pharmaceutical research field providing the validation that clinically effective non-viral nucleic acid therapeutics can be successfully developed [88,89]. Finally, the claim of the most promising drug delivery system was undeniably endorsed in 2020 when mRNA COVID-19 vaccines were developed and approved, even if issued through an emergency use authorization [90].

The key of the success of lipid-based delivery systems is the combination of the high drug loading capacity of both hydrophilic and hydrophobic drugs, as well as the possibility of being easily engineered to yield a desired size, surface charge, composition, and morphology just by modifying the phospholipid composition [91,92,93]. This last aspect is especially appealing when physical, biochemical, and cellular barriers hinder drug transport to its target inside the human body, as for the inhalation route. To this regard, another advantage of lipid-based NCs for inhalation lies in the fact that the most commonly employed lipid materials (i.e., phospholipids and cholesterol) constitute a significant portion of the naturally occurring pulmonary surfactant, which is the first barrier to get in contact with the inhaled particles [32]. Historically, liposomes were suggested as surfactants in patients with respiratory distress and, recently, a mixture of phospholipids has been commercialized for prophylaxis against distress symptoms in neonates (Survanta, AbbVie) [29]. 

Liposomal formulations may improve tolerability, increase compliance by reducing the dosing frequency, enhance the penetration of biofilms, and support the treatment of intracellular infections [94]. Furthermore, the hydrophobic nature of lipids, especially of neutral lipids, reduces the absorption of the ubiquitous vapor onto particles during inhalation, limiting aggregation and adhesion phenomena [95]. If it is true that the pinnacle of the development of a pharmaceutical product is when it enters clinical evaluation and demonstrates a meaningful benefit to patients, the approval in 2018 of amikacin liposomes for inhalation (Arikayce) is the ultimate proof that lipid-based NCs might also be a successful pharmaceutical tool for pulmonary administration [96], opening the path towards novel lipid-based inhaled treatments. 

In general, lipid-based delivery systems include lipoplexes that electrostatically self-assemble with negatively charged nucleic acids, liposomes composed of a phospholipids bilayer, and solid-lipid nanoparticles (SLNs) with a solid lipid core matrix enclosed in a lipid monolayer (Figure 2) [97,98]. 

Among lipid-based carriers of interest in inhalation therapy, both research and industry attention has been focused on liposomes as drug carriers for the encapsulation of small-molecule drugs or large proteins. Meanwhile, lipoplexes and LNPs are commonly used for the encapsulation of large cargoes such as proteins and nucleic acids (e.g., DNA, mRNA, siRNA, etc.) [99,100]. The main in vitro/ex vivo/in vivo findings achieved with lipid-based carriers for drug delivery to the lungs are reported in Table 1.

### 4.1. Liposomes

Liposomes can be considered as the first generation of lipid NCs, and they made their successful entry into the market in 1995 with the approval of the PEGylated liposomal formulation Doxil^®^. Since then, liposomes have been successfully investigated as a strategy to formulate a wide spectrum of pharmaceuticals (i.e., anticancer, antimicrobial, and anaesthetic agents, and vaccines) not only for parenteral delivery, but also for oral, pulmonary, or topical delivery [111]. In particular, inhaled liposome-encapsulated drugs represent a very promising strategy for application in cancer and CF therapy [112,113]. 

ARIKAYCE^®^ is the first and only liposome suspension for inhalation approved by the FDA to treat lung diseases caused by the Mycobacterium avium complex (MAC), a type of nontuberculous mycobacteria (NTM). Amikacin (AMK) is entrapped in liposomes (0.2–0.3 µm) composed of neutral, biocompatible lipids (i.e., DPPC and cholesterol), and ARIKAYCE has been developed for administration via an electronic nebulizer (eFlow^®^). The path that leads to approval has not been straightforward and has seen, all at once, the exploitation of several indications that, up to now, have not known the same fortunate ending [111]. From a general technological standpoint, liposome design is highly versatile, since single lipid blocks can be assembled in order to tune physicochemical properties and, consequently, optimize interactions with the lung environment, mucus, biofilm matrix, and bacterial cell surface [112]. In the specific case of CF, several studies have been conducted in order to better correlate the liposome composition with the in vivo performance, in term of stability, drug entrapment efficiency, drug release, as well as the ability to interact with the biological environment (i.e., different strain of *P. aeruginosa*), demonstrating how the composition plays a crucial role in the carrier design [114,115]. Furthermore, the therapeutic efficacy has been extensively investigated and the safety, pharmacokinetic advantage, and therapeutic effect of liposomes has been demonstrated in preclinical in vitro and in vivo models for antibiotics, such as tobramycin [116] as well as for chemotherapeutics molecules such as doxorubicin, gemcitabine, and paclitaxel [117,118,119]. In all cases, inhaled liposomes increase the drug retention, thus enhancing the therapeutic activity while simultaneously reducing the extra pulmonary side effects. Conversely, the systemic administration of liposomes resulted in a short residence time in the blood due to elimination via the reticuloendothelial system, which strongly limits their therapeutic application.

With an approach analogue to Arikayce, Aradigm developed liposome-based formulations for the lung delivery of ciprofloxacin: Pulmaquin^®^ (ARD-3150) to treat Infections in Non-Cystic Fibrosis Bronchiectasis (NCFB), Lipoquin^®^ (ARD-3100) to treat infections in CF patients, and, more recently, ARD-1100 for the local treatment and prevention of inhalation anthrax. Pulmaquin^®^ is a simple 1:1 mixture of Lipoquin^®^ (50 mg/mL) and free ciprofloxacin (20 mg/mL). In a Phase 2b study (ORBIT-2 and ORBIT-1), it showed superior performance as compared to Lipoquin^®^ alone. Therefore, Pulmaquin^®^ progressed into Phase 3 clinical trials in BE [120]. Unfortunately, at the end of two different phase 3 studies (ORBIT-3 and ORBIT 4), the efficacy of the inhaled ciprofloxacin agents in the treatment of patients with NCFB was controversial. Further research was required by the FDA, though Savara Inc. discontinued the work on Pulmaquin^®^ in December 2020 [121,122].

Novel inhalable and controlled release powder formulations of ciprofloxacin nanocrystals inside liposomes (CNL) were recently developed [123,124,125,126,127]. Though current data on the efficacy of inhaled liposomal antibiotics are quite encouraging, the use of inhaled liposomes is, in general, challenged by their well-established physical and chemical instability in aqueous dispersions for long-term storage, often causing vesicle aggregation, drug leakage, phospholipid hydrolysis, and/or oxidation and vesicle fragmentation during aerosolization via nebulizers [112]. To address these limitations, many methods have been investigated. For instance, the design of specially customized vibrating mesh nebulizers with larger mesh apertures that could have a less disruptive effect has been taken into account [128]. It was also shown how the composition could play a key role and that the use of cholesterol-enriched dipalmitoyl phosphatidylcholine or surface modifications could improve stability during nebulization [129,130]. Last but not least, liposomal dry powders for inhalation (e.g., lyophilizing, spray drying, and supercritical fluid technology) have been developed, showing suited features for lung deposition [131,132].

### 4.2. Solid Lipid Nanoparticles (SLNs)

Solid lipid nanoparticles (SLNs) have been investigated as a viable alternative to liposomes for drug and gene delivery to the lung. SLNs are characterized by a hydrophobic core assuring the ability to encapsulate both hydrophilic and hydrophobic therapeutics while remaining solid at 37 °C, assuring stability in vivo [122]. Compared with liposomes, SLNs offer improved physical stability before and after nebulization [104,133], a controlled release that can be modulated according to the environmental pH [105], and the easy industrial scale-up of the production techniques [134]. Moreover, the SLNs efficacy has been demonstrated in vivo upon nebulization [105,133]. On the other hand, low drug loading and drug expulsion during storage are the main disadvantages [135]. 

Nanostructured Lipid Carriers (NLCs) can be considered a “second generation” of SLN, and consist, at room and body temperature, of a liquid lipid matrix surrounded by a solid lipid shell. NLCs were developed to overcome the limitations faced by the SLNs, thus they are generally characterized by a higher encapsulation efficiency and finer control of drug release, and simple and inexpensive production on a large scale. In particular, the production aspect is a great advantage compared to liposomes [136,137]. 

SLNs and NLCs have been studied as a drug delivery system for various applications and for different administration routes [138,139]. With special regard to the lung administration, several advantages have been shown. Their small size helps the delivery and deposition to the lower respiratory tract with a prolonged residence time thanks to the ability to escape the evade clearance operated by alveolar macrophage. Furthermore, thanks to the lipophilic nature, they have shown optimal bio-adhesive properties [140].

Promising preclinical studies have been shown for the encapsulation of many drugs, such as anti-inflammatory [141,142], antibiotics [143,144], chemotherapeutic [145], and gene delivery, also in combination therapy [146,147]. In most cases, the SLNs and NLCs are formulated to be delivered through aerosolization, but also some dry powder for inhalation have been developed [106].

### 4.3. Nucleic Acid Delivery through Engineered Lipid Nanocarriers

In recent years, the use of lipid NCs encapsulating nucleic acids (NA) for the treatment of severe lung diseases has been gaining increasing attention. To this purpose, cationic lipid nanoparticles, or “lipoplexes”, and lipid nanoparticles (LNPs) are likely the most interesting ones [122,148].

Over the past few decades, cationic lipid nanoparticles have been the gold standard for NA delivery, taking advantage of the electrostatic interactions between negatively charged NAs and a cationic lipid, thus obtaining the so-called “lipoplex”, able to facilitate the interaction with the negatively charged cell membrane [149]. Felgner et al. [150] were the first to demonstrate the feasibility of the development of a lipid-based carrier by using a non-natural cationic lipid [N-[1-(2,3-dioleyloxy)propyl]-N,N,N trimethylammonium chloride (DOTMA)] to deliver plasmid DNA into eukaryotic cells lines. Since then, other cationic lipids commonly used in the production of lipoplexes have been 1,2-Dioleoyl-3-trimethylammoniumpropane (DOTAP), or the more advanced 1,2-dioleoyl-sn-glycerol-3-phosphoethanolamine (DOPE), which is especially used to improve the in vivo delivery. Unfortunately, lipoplexes bear the potential for inducing dose-dependent cellular toxicity [151]. Another important limitation to develop successful lipoplexes is the possible interaction between cation-charged lipids and the negatively charged region present in the CF mucus, resulting in the disassembling of the drug delivery systems. The inclusion of a third component on the surface, such as a PEG layer, has been proposed to stabilize the lipoplexes, demonstrating good mucus-penetrating properties [152].

The most ambitious nonviral clinical trial to date, involving cationic lipids, was conducted by Alton and colleagues. A CF transmembrane conductance regulator (CFTR) plasmid (pGM169) was formulated with Genzyme lipid 67 (GL67). The incorporation of small amounts of DMPE-PEG5000 enabled the preparation of concentrated lipoplexes with an optimal cationic lipid:pDNA ratio of 0.75:1 for aerosolization. Thus, a phase I clinical trial was initiated in 2008 followed by phase II clinical trials by the UK CF consortium (www.cfgenetherapy.org.uk, accessed on 26 February 2024). Patients received the nebulized lipoplex once per month for 1 year. Lung function was modestly stabilized in some individuals, and no significant adverse effects were observed. However, despite these encouraging results, the approach was not enough to achieve clear phenotypic correction [153,154].

In recent years, LNPs have emerged as a promising platform for RNA delivery and have shed light by resolving the inherent instability issues of naked RNA, thereby enhancing the therapeutic potency. LNPs consisting of ionizable lipids, helper lipids, cholesterol, and poly(ethylene glycol)-anchored lipids can stably enclose RNA and help them release into the cells’ cytosol [155]. The approach that is leading the path to overcome all the limitations imposed by cationic lipids is the substitution of the quaternary ammonium head with a titratable moiety which produces an ionizable lipid. Pieter Cullis’ research group has been the first to exploit the potential associated with the use of an ionizable lipid in order to deliver nucleic acids, and their studies have opened the way to, first, the approval of Onpattro^®^ in 2018 and then mRNA vaccines for COVID-19. Having an ionizable excipient offers the possibility of changing the charge status according to the environmental pH, and in this way, it is possible to maximize the interaction with the nucleic acid during the production phase, have a stable complex in the bloodstream (or at physiological pH), and finally, offer an effective escaping solution once the vector is inside the acidic pH of the endosome thanks to the re-ionization of the amino lipid component and the formation of electrostatic and hydrophobic interactions between disassembled lipids and the endosomal membrane [156,157,158]. 

Onpattro^®^ approval represents a milestone to many extents. From a therapeutical point of view, it gives new hope to the hATTR patients who can now count on an approach able to stop the progression of the disease. From an siRNA development point of view, Onpattro^®^ is not only the first non-viral vector that made it to the market, but it also opened a new perspective for the development of a more universal option to solve the endosomal escape problem. Furthermore, we cannot forget that based on the proof of concept provided by Onpattro^®^ technology, the COVID-19 vaccines were developed and were, for the first time, administrated on a large scale [159]. LNPs’ ability to deliver mRNA to the inside of cells is not only limited to vaccination, but has versatile applications such as treating genetic disorders [160,161]. Recently, LNPs were employed in a clinical trial to deliver Cas9 mRNA and guide RNA for editing the gene causing transthyretin amyloidosis [162]. The therapeutic application of the LNP platform is restricted mostly for hepatic diseases because LNPs innately accumulate in the liver when administered systemically, which significantly limits its access to other organs [88]. However, investigations into applying LNPs to deliver inhaled therapeutics to the lungs are underway [122]. In fact, even if recent studies have shown that modulating the nanoparticle surface charge permits the systemically administered LNPs to reach the lungs [27,163,164,165], the focused delivery of nucleic acids-based therapeutics to the lungs via the inhalation route still represents the most promising approach to treat severe lung diseases, providing stronger control over the induction of off-target effects. The designing criteria of LNPs are under evaluation in order to adapt LNPs to inhalation [97,109]. 

Encouraging results have already been shown in terms of aerodynamic properties for deposition to the lower respiratory tract, with good stability upon nebulization [97,109] and in vivo activity through intratracheal administration [110]. Moreover, the feasibility of engineering LNP-based powders by spray drying was recently demonstrated [137,166]. Optimized spray-dried LNPs penetrated the lung mucus layer and maintained bioactivity resulting in >90% protein downregulation with a confirmed safety profile in a lung adenocarcinoma cell line. Furthermore, the spray-dried LNPs successfully achieved up to 50% gene silencing of the house keeping gene GAPDH in ex vivo human precision-cut lung slices without inducing any toxic effect, as shown by the cytokine levels [109]. 

## 5. Polymer-Based Nanocarriers for Lung Administration: State-of-the-Art

In the last decade, polymeric NCs have gained considerable interest regarding their pulmonary delivery applications due to their unique properties [79,167,168,169]. They represent a well-established platform for the encapsulation and delivery of a multitude of therapeutic molecules due to their versatility in polymer physiochemical properties as well as the variety of available production techniques, which can be selected in view of the specific drug and intended application [6,170,171]. The most employed polymers in NC design and development are biocompatible, biodegradable, and capable of the sustaining the release of the encapsulated drug without the relevant side effects or cytotoxicity. Furthermore, recent studies underline how polymeric NCs may assist drug diffusion across the lung barriers, which are paramount for efficient drug delivery in the lungs [79,172]. To date, several materials have been studied to achieve optimal polymer NCs for inhalation. Depending on the polymer properties and the adopted production technique, various polymeric NCs can be obtained, such as nanogels, nanospheres, and polyplexes (Figure 3) [2,173].

According to their nature, the systems can be broadly classified in natural and synthetic polymer-based NCs. 

### 5.1. Natural Polymer-Based Nanocarriers 

Natural polymers appear to be very interesting materials in NC production thanks to their biocompatibility, low toxicity, and biodegradability [174]. Furthermore, the techniques usually employed in the production of natural polymer-based NCs (i.e., crosslinking gelation) are very gentle and characterized by low shear forces, thus they are ideal for the encapsulation of unstable molecules [175,176]. The main in vitro/in vivo findings achieved with natural polymer NCs for drug delivery to the lungs are reported in Table 2. 

Between the natural materials for pulmonary delivery, albumin is one of the most studied polymers due to its low antigenicity, low toxicity, biocompatibility, biodegradability, low costs, and abundance. Serum albumin is the most abundant protein in the plasma, and it is characterized by a high affinity with different molecules; thus, as a nanocarrier, it has been chosen to incorporate a variety of active compounds [177,178,179].

**Table 2 pharmaceutics-16-00347-t002:** Main in vitro/in vivo findings achieved with natural polymeric NCs for the pulmonary delivery of drugs.

Polymer	Encapsulated Molecule	In Vitro Model	In Vivo Model	Main Findings	Ref.
Albumin	Tacrolimus	-	Intratracheal administration in bleomycin-induced pulmonary fibrosis mouse	Anti-fibrotic effect significantly higher than intraperitoneal administration	[180]
Albumin	-	Macrophages derived from BALB/C mice	Oropharyngeal aspiration in male BALB/C mice	High in vivo biocompatibility with mild inflammation at highest dose tested. Slower clearance. No accumulation in major organs	[174]
HSA	Benzothiazinone (BTZ043)	Murine bone marrow-derived macrophages infected with *M. tuberculosis*	Intranasal instillation in old female C3HeB/FeJ mice infected with *M. tuberculosis*	Enhanced efficacy in vitro compared to the free drug; reduced bacterial load in vivo	[181]
TRAIL-HSA	Doxorubicin	Apoptotic and cytotoxicity activity on H226 cell line (human lung squamous carcinoma cell line)	Insufflation of nanoparticle dispersion in mouse bearing H226 cell-induced metastatic tumors	Synergistic apoptotic activity and anti-tumor efficacy in vitro and in vivo	[182]
BSA	siRNA	Cellular uptake and cytotoxicity on A549 cell line; gene-silencing on KRAS G12S mutant A459 cells line	-	Low cytotoxicity with enhanced cellular uptake. High knock-down efficiency in vitro	[183]
CS	Influenza vaccine	Cytokines secretion in porcine monocyte-derived dendritic cells	Intranasal nebulization in pigs	Augmented cross-reactive T and B lymphocytes response	[184]
CS	Bedaquiline	Cytotoxicity profile on macrophage cell line	Inhalation of freeze-dried nanoparticles in rats	Low acute and chronic toxicity in vivo	[185]
CS	Salmon Calcitonin	-	Intratracheal administration in rats	Higher absorption and deposition in deep lung	[186]
CS	Prothionamide	-	Intratracheal administration of dry powder containing nanoparticles in rats	Prolonged drug persistence in lungs	[187]
CS-HA	Gallium (III)	Human epithelial bronchial cells (16HBE14o-) and *P. aeruginosa*	Intratracheal administration of dry powder containing nanoparticles in rats	Improved accumulation of drug in lung tissue and high tolerability in vivo	[188]
CS-PVA	Magnolol	Cytotoxicity profile on cells A549 cell line	-	Enhanced lung deposition with high cell viability	[189]
ALG_CS-DNase	Tobramycin	Antimicrobial activity on CF sputum sample and *P. aeruginosa* strain (PA01)	Injection of nanoparticles dispersion in *Galleria melonella*	Increased penetration across CF sputum and enhanced anti-pseudomonal activity in vitro and in vivo	[190]
ALG-CS/Tween80	Rifampicin and ascorbic acid	Antibacterial activity on *Mycobacterium* *Tuberculosis* (M. tb.); cytotoxicity on kidney epithelial cells	-	Increased antibacterial activity Low cytotoxicity on kidney epithelial cell lines	[191]

List of Abbreviations: CS: Chitosan; BSA: Bovine-serum albumin; PVA: Poly(vinyl alcohol); siRNA: small interfering RNA; ALG: alginate; GCS: glycol chitosan; TGA: thioglycolic acid; HSA: human serum albumin; TRAIL: tumor necrosis factor (TNF)-related apoptosis-inducing ligand.

In recent years, albumin NCs have gained considerable research attention as a drug delivery system owing to the approval by the FDA of nanoparticle albumin-bound (NAB) paclitaxel (Abraxane^®^) in the treatment of metastatic breast cancer (2005), advanced/metastatic non-small cell lung cancer (2012), and metastatic pancreatic cancer (2013). Inspired by the success of Abraxane^®^, albumin-based NCs have stimulated interest also for inhalation [174,181,182,192,193]. The first in vivo proof-of-concept study on the lung biocompatibility and biodistribution of inhaled albumin NCs was performed by Woods et al. [174]. The results showed the absence of a significant inflammatory response in mice after the single pulmonary administration of bovine serum albumin (BSA) NCs as compared to the control BSA solution. Meanwhile, SPECT/CT imaging and post-mortem organ biodistribution studies demonstrated that lung tissue accumulation up to 48 h was significantly higher for BSA NCs compared with the control BSA solution. The absence of major NCs accumulation in secondary organs, and likely of related side effects, was further encouraging. In view of the proven efficacy of serum albumin NCs for cancer therapy [177], their potential for the direct delivery of anti-cancer drugs by inhalation has been explored [182]. Briefly, human serum albumin (HSA) NCs were loaded with doxorubicin and modified with the apoptotic protein TRAIL (tumor necrosis factor-related apoptosis-inducing ligand) to maximize specificity for lung cancer cells. The resulting TRAIL/Dox HSA-NCs exhibited a robust anti-tumor effect after pulmonary administration in a xenograft mouse model [182]. Due to the presence of charged amino acids able to electrostatically interact with charged molecules, albumin raised awareness also for the delivery of nucleic acid therapeutics. With this idea in mind, Merkel and co-workers developed BSA NCs for the delivery of siRNA targeting the KRAS G12S mutation, the most frequent mutation in human cancers. The study demonstrated that BSA NCs could protect the siRNA payload against RNases, enable in vitro transport to A549 cells, and mediate significant sequence-specific KRAS knockdown, resulting in the reduced cell growth of siRNA-transfected lung cancer cells [183]. 

A relevant issue in the development of inhalable NCs based on albumin is represented by unfavorable mucus/carrier interactions. In fact, albumin may establish quite strong interactions with mucus components (i.e., an NC/mucin electrostatic interaction or covalent binding to mucin), which could be mitigated incorporating (e.g., cyclodextrin) or surface-engineering (e.g., PEGylation) polymers in NCs’ design [194,195]. The mucoadhesivity of albumin NCs prompted the exploration of their potential for local rather than systemic delivery through the lungs. Along this line, J. Seo and co-workers demonstrated the superior efficacy of inhaled tacrolimus-bound albumin NCs in mice with bleomycin-induced pulmonary fibrosis as compared to an intraperitoneal tacrolimus solution [196]. When conceiving these systems for pulmonary delivery, the results of Papay et al. are also relevant, as they successfully developed lactose-based dry powders for inhalation embedding apigenin-loaded albumin NCs, likely useful for the local treatment of lung injury in asthmatic conditions [197]. 

Among natural polymers, chitosan (CS) has been widely investigated for pulmonary drug delivery [198].

The positive charge of CS is a critical attribute to improve the absorption by opening the tight junctions of the lung epithelium [199,200]. Furthermore, CS charges are responsible for the interaction with mucus components, and thus for the mucoadhesive effect, and the amount of charges could be tuned through the deacetylation grade [201]. In recent years, numerous studies have indicated CS-based NCs as promising inhalable delivery systems [202], both for systemic and local delivery. CS NCs are usually prepared by the gelation technique, coacervation, nanoprecipitation, and the reverse micellar method [203]. They are effective in protecting, controlling the release, and promoting drug absorption through lung epithelia when administered both as NC dispersion and embedded into inert carrier microparticles (Nano-embedded Microparticles—NEM) [186,189,204].

One of the most studied and interesting applications of CS is the development of drug delivery systems (DDS) for the pulmonary administration of antimicrobial drugs [185,205]. In fact, CS itself shows antibacterial effects likely ascribable to the electrostatic interactions between its protonated amino groups and the phosphoryl groups and lipopolysaccharides of bacterial cell membranes [206,207,208]. The consequent membrane perturbation and release of the cell content may result in an increased therapeutic effect of the drug cargo [209]. Furthermore, the affinity of CS with alveolar macrophages, due to the electrostatic interactions between the positively charged polymer and the negatively charged sialic acid on the cell membranes [210], boosted the development of CS-based delivery systems for the local treatment of tuberculosis. Different drugs have been encapsulated into CS or CS-derivative NCs with increased anti-tubercular activity with respect to the free drugs [185,211,212].

The mucoadhesive tendency of CS (mostly due to its chemical structure) is both an advantage and a disadvantage and has solicited the researchers’ attention. In this direction, even if inhalable CS NCs were shown to be effective in different in vivo applications [188,201,213], it was recently highlighted how CS-NCs may be entrapped in the superficial mucus layer, which is quickly cleared, and are not expected to reach the underlying airway epithelium [214,215]. As a matter of fact, the current trend is based on the development of NCs modified on the surface with inert polymers, which can improve the particle mobility into the mucus gel layer [26,216,217]. Another issue is the in vivo elimination of inhaled CS NCs. Though CS NCs display an acceptable safety profile and low chronic toxicity [185], no study has conclusively demonstrated the complete biodegradation or elimination of CS NCs in vivo. Nevertheless, this aspect has been reviewed to help shed light on this point [218].

### 5.2. Synthetic Polymer-Based Nanocarriers

The pulmonary route has some distinctive features that significantly limit the panel of synthetic materials available for inhalation. Biodegradable aliphatic polyesters, such as poly (lactic-co-glycolic acid) (PLGA) or Poly(N-isopropyl acrylamide)(PNIPAM), are the most exploited polymers because of their biodegradability, biocompatibility, and versatility [2,219,220,221]. The properties of the polymer, such as the grade of hydrophobicity and the degradation rate, and those of the resulting NCs, such as the particle size, surface properties, and drug encapsulation/release, can be easily tuned in order to optimize the affinity and the compatibility of the delivery system with the lung environment. 

The main in vitro/in vivo findings achieved with synthetic polymer NCs for drug delivery to the lungs are reported in Table 3.

As reported in Table 2, PLGA is among the polymers that have been extensively explored for the development of inhalable NCs. If PLGA-based NCs are considered promising in terms of drug encapsulation, protection, controlled release, and aerodynamic properties, these systems appear not always efficient in crossing cellular/extracellular pulmonary barriers. As a consequence, the examples of inhalable PLGA NCs reported in the literature are often engineered at the surface with hydrophilic polymers, which can shield nanocarrier interactions with lung-lining fluids, thus promoting drug transport to the target [222,223,224,231,232,233]. 

Polyethylene glycol (PEG) is the most widely exploited polymer to achieve mucoinert PLGA-based NCs. In fact, the PEGylation leads to the formation of a hydrophilic shell and neutral surface charge that prevents hydrophobic or electrostatic NC interactions with the mucus gel layer [26,234,235]. Surface properties can be further tuned by changing the PEG molecular weight and its density on the particle surface [234,235]. The transport of PEGylated NCs to the inner layer of the mucus likely increases their retention in the lungs by reducing the clearance mechanism and facilitating macrophages’ escape [26,214,230]. The effectiveness of PEGylation in the pulmonary drug delivery of PLGA NCs was extensively explored and promising results have been achieved in both in vitro and in vivo models in different therapeutic applications, such as antimicrobial and anticancer therapy [236,237].

Alternative polymers for the functionalization of NCs able to assist their penetration through mucus have been investigated. These include poly(2-oxazolines), polysarcosine, zwitterionic polymers (polybetaines), proteolytic enzymes, and poly(vinyl alcohol) (PVA) [28]. Between them, PVA is the most employed emulsifier in polymeric nano- and microparticles and it is able to provide uniform particulate systems with low polydispersity. Recently, we have developed PLGA NCs engineered with PVA for the delivery of an antimicrobial peptide, Esculentin, to the lung [224]. We have found that the PVA shell can assist the particles transport across the mucus layer, which is very low for the naked antimicrobial peptide. Furthermore, the NCs can control the release of the encapsulated molecule, providing a prolonged antimicrobial effect on *Pseudomonas aeruginosa* both in vitro, on bacteria culture, and after pulmonary administration as NC dispersion in an acute infected murine model in vivo. Although the PVA is generally considered a mucoadhesive polymer, the ability of PVA to improve the particle diffusion across the mucus was found to be related to the grade of hydrolysis, the molecular weight, and the density of the particle shell. In particular, the non-covalent shell of partially hydrolized PVA, with a hydrolysis degree between 75 and 95%, were found to improve the PLA NCs’ mobility in mucus [238].

Despite the efficacy in overcoming the mucus barrier, NC engineering with hydrophilic polymers seems to not be the best strategy in cellular target-based therapies, such as gene delivery, in which the NCs need to be designed for cell penetration [239].

An early exploited strategy to achieve mucus-penetrating particles able to improve the drug cell uptake is represented by the hybrid lipid-polymer NCs [55,172,240]. Hybrid NCs consisting of a PLGA core and a lipid shell of dipalmitoylphosphatidylcholine (DPPC) have been developed in our labs for the pulmonary delivery of siRNA. Thanks to its ability to form a shell around the polymer core, DPPC was chosen to improve NC compatibility and tolerance in the lung environment. Furthermore, it has been demonstrated that a DPPC shell can provide mucoinert and muco-diffusive properties to the particulate system while improving the drug cell uptake, crucial in siRNA therapies [46,74,163,230]. This strategy appears very promising in gene delivery, in which different lipids were used with success for the development of inhalable hybrid NCs [240,241]. In order to assist the gene uptake, particle surface engineering is often associated with the encapsulation of non-viral vectors as cationic polymers. One of the most employed in gene delivery is polyethylenimine (PEI), which can electrostatically interact with nucleic acids forming complexes (polyplexes), improving the encapsulation and release from polymeric NCs and facilitating gene cellular internalization [55,230]. A recent study investigated the impact of various biomimetic endogenous pulmonary phospholipids on the in vivo behavior of PLGA NCs. The study revealed that surface engineering with neutral phospholipids, such as DPPC and dipalmitoylphosphatidylamine (DPPE), resulted in a reduction in the uptake of NCs by alveolar macrophages, while the use of negatively charged phospholipids, such as dipalmitoylphos-phatidylserine (DPPS) and dipalmitoylphosphatidylglycerol (DPPG), enhanced the uptake. Regardless of the charge, phospholipid surface engineering increased the uptake of NCs by A549 cells and drug retention in the entire lung. DPPC, DPPE, and DPPG facilitated drug retention in BALF, whereas DPPS facilitated drug absorption in lung tissue. However, no impact of phospholipids on drug tissue distribution was observed [242]. Concerning the particle size, NCs with an hydrodynamic diameter of 100 nm showed significantly higher lung retention and tissue adsorption than NCs with a higher diameter (300, 800, and 2000 nm); therefore, NCs with a small size are the most effective for pulmonary drug delivery, especially for local lung disease therapy [243].

In order to conceive mucus-penetrating and long-residence DDS for pulmonary administration, macrophage uptake must be considered. Sometimes, being a further limit in drug pulmonary administration, it becomes a desired effect in specific macrophage-target therapy. In fact, the selective and controlled internalization of PLGA-based NCs loaded with anti-infective drugs by alveolar macrophages is under investigation for targeted antimicrobial therapy [7,183,210]. This is the case of antitubercular therapy, in which the target is *Mycobacterium tuberculosis* located in the alveolar macrophages (host cells) [244]. Unmodified PLGA NCs, which are usually inappropriate for pulmonary delivery due to their fast elimination by macrophage clearance, have been successfully developed as inhalable systems for the delivery of three frontline antitubercular drugs [245]. In the attempt to improve the NC uptake in macrophages and thus increase the drug concentration at the site of action, appropriate surface modifications of PLGA NCs have been explored. A recent study underlined the higher effectiveness of PLGA nanocapsules, modified with CS, in killing intracellular *Staphylococcusaureus* and *Micobacterum abscessus* with the same dose of the drug in its free form. The CS confers to NCs with a positively charged surface which is able to increase the uptake in macrophages and improve the drug effect [223]. 

Surface engineering the NC with CS has been exploited in different applications, such as lung infections and pulmonary fibrosis. As a matter of fact, despite the controversial discussion about the higher effectiveness of muco-inert particles with respect to the mucoadhesive ones, CS-modified NCs are able to improve the drug in vivo bioavailability after inhalation [180,222]. The CS mucoadhesive effect can increase the NC residence time in the lung and, thanks to the opening effect on the tight junction, can increase the drug absorption, leading to high systemic bioavailability. Furthermore, thanks to the antimicrobial effect, CS has gained great interest in antimicrobial therapies in association with PLGA in order to achieve NCs able to control drug encapsulation and release (PLGA core) while tuning the particle/infected lung environment interactions (CS coating). One of the main barriers in bacterial infections in the lung is represented by the bacteria biofilm, which some Gram (-) bacteria can produce. In particular, the ability of CS-covered PLGA NCs to effectively penetrate the bacteria biofilm of *Pseudomonas aerugionosa*, providing an antimicrobial depot in situ and to enhance drug activity for a longer time, has been demonstrated [246,247].

Just like the lipid nanoparticles mentioned earlier, numerous polymeric nucleic acid carriers form electrostatic interactions to bind their payload. These carriers are classified into two types—the polyelectrolyte complex (polyplex) and the polyplex micelle (micelleplex). The micelleplex formation typically incorporates hydrophobic groups to stabilize the polyplexes, which would otherwise rely solely on electrostatic interactions for their structural stability [248].

The formation of both poly- and micelleplexes depends on electrostatic interactions, which are established via protonable amine groups. As a result, polymeric materials with polyamine groups, such as polyethyleneimine (PEI), polylysine (PLL), or poly-(amidoamine) PAMAM, have been extensively studied for nucleic acid delivery [249]. For instance, PEI-based polyplexes have demonstrated significant potential in pulmonary delivery, and their ease of modification enables the attachment of shielding agents, targeting moieties, or lytic peptides to enhance their efficacy [250,251,252].

However, due to the inherent toxicity often associated with polyamines, it became clear that alternatives are required to develop safe and efficient NCs using polyplex systems. Poly(beta-amino esters) (PBAEs) emerged as a popular alternative due to their favorable toxicity profile. Several studies have demonstrated impressive results with inhaled PBAE formulations in efficiently condensing mRNA, facilitating intracellular uptake and releasing the gene cargo at the cytosol level to allow the translation of the encoded protein [252]. 

Furthermore, poloxamines have also been shown to effectively deliver RNA and DNA cargos to the lungs in CF treatment [212]. The developed NCs have demonstrated that they are able to facilitate the long-term restoration of CFTR in CFBE-delF cells and CF mice with a favorable safety profile.

Despite the great potential of pulmonary nanotherapies, no products based on polymeric NCs, neither natural nor synthetic, are approved for human use. The complete evaluation of NC toxicity in vivo after aerosolization remains a significant challenge. Numerous are the results achieved in recent years, but an additional effort is needed to assess the safety of inhalable nanoformulations. 

## 6. Harnessing Nanocarriers for Inhalation: From Liquid Aerosols to Dry Powders

In order to square the circle of particle deposition in the deep airways, an approach that is gaining success is the development of nano-embedded microparticles (NEM). NEM are usually produced by spray drying or freeze drying and consist of nanocomposite particles obtained by the inclusion of drug-loaded NCs within microparticles made of an inert material (i.e., lactose, mannitol) [227,253]. The underlying concept is that, once the inert carrier reaches the deep lung and dissolves in the lung-lining fluid, the primary NCs are released to exert their action. The addition of this second level of complexity is expected to add a number of advantages to those already arising from the presence of an NC system, for instance, all the advantages arising from the micro-sized particles, such as better flow and aerosolization properties resulting in the engineering of a potentiated system for local therapy [16,254]. Furthermore, through a careful selection of the carrier material, it is also possible to further functionalize the delivery system. For instance, we have shown how hydroxypropyl beta cyclodextrin (HPβCD) can contribute to the inhibition of the biofilm activity in lung infections triggered by *P. aeruginosa*. This effect, based on previous evidence, can be explained by the entrapment of N-acyl homoserine lactones, involved in bacterial quorum sensing inside the CD cavity [84,255]. In another work, the addition of mannitol, thanks to its innate mucolytic activity, assists the released NCs in the navigation of the biofilm barrier, one of the biggest challenges in the development of CF therapies for lung administration [254,256]. Of note, to realize the full potential of this approach is necessary to have a thorough understanding of the physiological barrier and of the modifications that happen in pathologic conditions [172]. From a translational perspective, if the right attention is devoted to the process parameters, a powder can be manufactured which is ready to use as an MDI or DPI formulation featuring long-term stability, often even at room temperature [109,257]. However, a drawback that needs to be considered is the unavoidable dilution caused by the additional excipient that impacts the concentration of the active pharmaceutical ingredient (API), leading to a higher dose due to the diluted drug content, potentially raising concerns about tolerance as well [258]. This underscores the significance of carefully selecting the most suitable excipient. Some authors suggest that the choice of an excipient should go beyond serving as a stabilizer; it should also function as a shell former during the spray-drying process to optimize drug loading in the formulation [257].

## 7. Patents on “Inhalable Nanocarriers”

The rising interest in the application of NCs in inhalable therapy has promoted the patent applications. The most relevant patents in this field are reported in Table 4, as identified by searching on the Espace Patent search web site (https://worldwide.espacenet.com, accessed on 23 January 2024) “Inhalable nanocarriers”. 

## 8. Conclusions and Perspectives

NCs have emerged as a promising strategy for drug delivery to the lungs, showcasing their potential in overcoming biological barriers, such as mucus and, in the context of pulmonary infections, the bacterial biofilms. The unique properties of NCs, including their size, surface charge, and tailored surface functionalities, enable them to navigate through the complex pulmonary environment. Their nanoscale dimensions facilitate penetration through mucus layers, while surface modifications can enhance interactions with specific cell target. Moreover, NCs offer controlled drug release profiles, improving drug efficacy and minimizing potential side effects. These characteristics make NCs a valuable strategy for targeted drug delivery to the lungs, particularly in respiratory diseases where overcoming lung barriers is pivotal for therapeutic success. The unique properties of NCs allow for the targeted delivery of drugs to the lungs, improved drug diffusion across the lung barriers, and the controlled release of drugs. However, several challenges still need to be addressed to fully realize the potential of NC-based therapies for pulmonary drug delivery.

One major challenge is the potential toxicity of NCs to lung cells, which requires the careful evaluation and optimization of the NC properties. Additionally, there is a need for further research to optimize the NC formulation, stability, and pharmacokinetics in vivo. The use of advanced techniques for particle production and characterization can help to appropriately outline the fate and behavior of NCs in the lungs.

Despite these challenges, NC-based pulmonary drug delivery has the potential to revolutionize the treatment of respiratory diseases, including gene therapy, lung infections, and rare diseases (i.e., CF). Continued research and development in this field can lead to the development of more effective and targeted therapies with fewer side effects, improving patient outcomes and quality of life. 

## Figures and Tables

**Figure 1 pharmaceutics-16-00347-f001:**
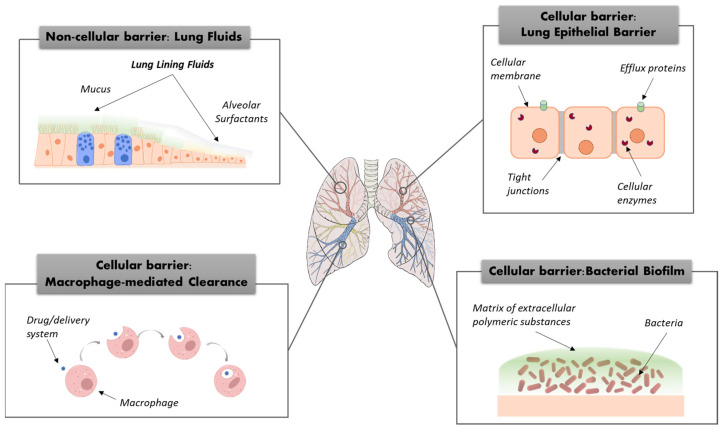
Schematic diagram showing the barriers imposed by the lung to inhaled drugs and drug-loaded NCs.

**Figure 2 pharmaceutics-16-00347-f002:**
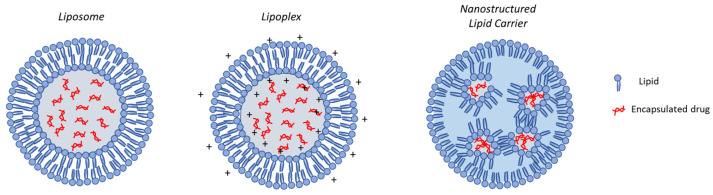
Schematic representation of different architectures of lipid-based NCs.

**Figure 3 pharmaceutics-16-00347-f003:**
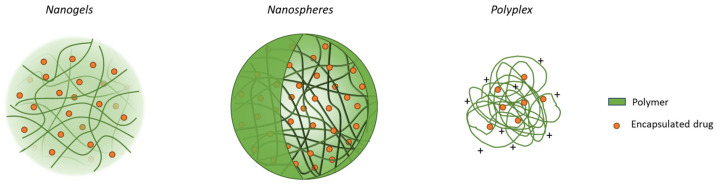
Schematic representation of different architectures of polymer-based NCs.

**Table 1 pharmaceutics-16-00347-t001:** Main in vitro/ex vivo/in vivo findings achieved with lipid NCs for the pulmonary delivery of drugs.

Lipids	Encapsulated Molecule	In Vitro Model	Ex Vivo/In Vivo Model	Main Findings	Ref.
Liposome DPPC; CHOL; DSPE-PEG2000	Methylprednisolone N-acetyl cysteine	In vitro treating LPS-stimulated RAW 264.7 macrophages	Accumulation and therapeutic efficacy in LPS-induced lung inflammation model induced on C57BL/6 mice	Decrease in TNFα and nitric oxide secretion in LPS-stimulated RAW 264.7 increased penetration through the mucus; increased accumulation in vivo over 48 h	[101]
FA-modified liposomes DSPC; DSPE-PEG2000-Folate	Rapamycin	Intracellular distribution, cellular association, and cytotoxic activity on KB, LL2, and A549 cells line	Intrapulmonary behavior and stability on Wistar rats; in vivo anti-tumor activity on male C57BL/6NCr mice	Better uptake through FR and autophagy-mediated cytotoxicity; good stability in BALF and longer survival upon pulmonary administration	[102]
CS-coated liposome DPPC; CHOL; HSPC; DPPG	Oxymatrine (OMT)	Cytotoxicity and inhibitory effects on HEp-2 cells	Biodistribution upon tracheal intubation in BALB/c female mice and treatments on RSV-infected mice	Enhanced distribution and retention of OMT in lung tissue alleviative effect of OMT on lethal RSV-infected mice	[103]
SLN Compritol 888	Favipiravir	Cytotoxicity Assay on Vero-E6 Cells	-	Promising activity of inhalable SLPs encapsulating FPV against coronavirus	[104]
SLN PP, ODA:	Prodrug of isoniazid	Antibiotic activity against free MSG and intracellular MSG; cytotoxicity on raw 264.7 and A549 cells	In vivo antibiotic efficacy on a Wistar rat model infected by MSG	Macrophage-targeting and pH-sensitive property	[105]
SLN lecithin, CHOL, PHC TopFluor^®^	siRNA	Cell transfection and determination of TNF-α expression in J774A.1 cell line	-	Aerosolizable dry powder by thin-film freeze-drying (TFFD)	[106]
SLN CP	Fluorescent probes as models	-	Ex vivo/In vivo fate of inhaled nanocarriers upon administration in BALB/c mice.	Positive correlation between particle size and lung retention time	[107]
SLN lecithin, CHOL, GA-PEG-SA conjugate	Gefitinib	Internalization and activity on A549	-	Increased cellular uptake and superior anticancer effect compared to free gefitinib	[108]
LNP CHOL; PEG-DMG, DSPC, DSPG; DOTAP; sulfur-containing analog of DLin-MC3-DMA	siRNA	In vitro cytotoxicity on H1299-GFP cells; in vitro GFP protein downregulation	Ex vivo activity of spray-dried LNPs in human precision-cut lung slices (hPCLS)	Long-term stable dry powder with good gene silencing efficiency	[109]
LNP DLin-MC3-DMA, PEG-DMG, DSPC, CHOL or β-Sitosterol	mRNA	In vitro transfection of LNP before and after nebulization on HeLa cells	Pulmonary transfection by LNP through a mouse nebulizer delivery system on BALB/c mice	Enhanced protein expression in vitro and in vivo without inducing toxicity	[97]
LNP DOPE; PEG-DMG; DMPE-PEG; DSPC; DPPC	mRNA	Intracellular Protein Expression in HEK-293 cells	In vivo transfection on BALB/c mice	Aerosolization-mediated pulmonary mRNA delivery and expression in vivo	[110]

List of Abbreviation: DPPC: 1,2-dipalmitoyl-sn-glycero-3-phosphocholine; CHOL: Cholesterol; DSPE PEG2000: (1,2-distearoyl-sn-glycero-3-phosphoethanolamine-N-[amino(polyethylene glycol)-2000]); LPS: lipopolysaccharide; TNFα: tumor necrosis factor alpha; DSPC: (1,2-distearoyl-sn-glycero-3-phosphocholine); DSPE-PEG2000-Folate: 1,2-distearoyl-sn-glycero-3-phosphoethanolamine-N-[folate (polyethylene glycol)−2000]; FR: folate receptor; FA: folic acid; HSPC: hydrogenated soybean phosphatidylcholine; DPPG: 1,2-dipalmitoyl-sn-glycero-3-phospho-(1′-rac-glycerol); PP: Palmityl Palmitate; ODA: octadecyl amine; MSG: Mycobacterium smegmatis; TopFluor^®^: 1-palmitoyl-2-(dipyrrometheneboron difluoride)undecanoyl-sn-glycero-3-phosphocholine; P2: aza-BODIPY-structured ACQ probes; P4: za-BODIPY analog of P2; DiR: a non-water-quenching probe; TFFD: thin-film spray drying; CP: Cetyl palmitate; PEG-DMG: 1,2-Dimyristoyl-rac-glycero-3-methoxypolyethylene glycol-2000; DSPG: 1,2-Distearoyl-sn-glycero-3-phosphoglycerol; DOTAP: 1,2-dioleoyl-3-trimethylammonium-propane (chloride salt); DLin-MC3-DMA 4: (dimethylamino)-butanoic acid, (10Z,13Z)-1-(9Z,12Z)-9,12-octadecadien-1-yl-10,13-nonadecadien-1-yl ester; DOPE: 1,2-dioleoyl-sn-glycero-3-phosphoethanolamine; DMPE-PEG: Polyethylene Glycol 1,2-Dimyristoyl-sn-Glycero-3-Phosphoethanolamine.

**Table 3 pharmaceutics-16-00347-t003:** Main in vitro/in vivo findings achieved with synthetic polymeric NCs for pulmonary drug delivery.

Polymer	Encapsulated Molecule	In Vitro Model	In Vivo Model	Main Findings	Ref.
PLGA and PLGA-CS	Voriconazole	-	Freeze-dried NCs aerosolization through a nose-only inhalation chamber in mouse	Prolonged retention time in the lung and plasma	[222]
PLGA and PLGA-CS	Clarithromycin	Cytotoxicity on Calu-3 and THP-1 cell lines; internalization in infected macrophage cell lines; permeability across Calu-3 cells grown at ALI	Aerosolization in murine and zebrafish *S. Aureus*-infected model	Risen permeability in vitro; reduced bacteria load in vivo	[223]
PLGA-PVA	Antimicrobial peptide (AMP) (Esculentin 1–21)	NC diffusion across AM and simulated biofilm. Antimicrobial activity of AMP-loaded NCs against strain *P. aeruginosa* ATCC 2785	Intratracheal instillation of NC dispersion in healthy and *P. aeruginosa*-infected mouse	Enhanced drug permeability and efficacy in vitro. Reduction of bacterial load in vivo	[224]
PLGA-PVA	Etionamide	-	Intratracheal aerosolization of freeze-dried NCs in rat	Increased drug persistence in the lung and reduced systemic absorption	[225]
PLGA-PLX-188	Docetaxel	Anti-cancer activity on A549 cell line	Intratracheal instillation of NC dispersion in rat	Enhanced and sustained cytotoxicity in cancer cells; high residence time in the lung	[226]
PEI-Mannitol/Threalose	DNA	Uptake and transfection efficiency on A549 cell line	-	Uptake and transfection profiles maintained after redispersion	[227]
BSA- PEI -PLGA	Quinacrine	Uptake, cytotoxicity, and clonogenic assay on A549 cell line. Anti-proliferative activity on A549 3D-Spheroid cell line	-	Improved anticancer activity with high uptake and efficient tumor penetration	[228]
PEG-GET	DNA	Uptake and transfection studies on BEAS2B-R cell line. Multiple particles tracking in CF patient mucus	Intratracheal aerosolization of NCs’ dispersion in healthy mouse	Maintained NC colloidal features in CF sputum. Increased in vivo safety profiles with high biodistribution and transgene expression	[229]
PEI-PLGA-DSPE_PEG	pDNA	Cell viability and uptake studies on CFBE41o-cell line. Permeation in mucus layer	Intratracheal administration of dried NC in non-pathogen rat	Improved transfection efficiency with high safety profiles and diffusion through the mucus layer	[230]

PLGA: poly(lactic-co-glycolic) acid; CS: Chitosan; PVA: polyvinyl alcohol; AM: Artificial Mucus; PLX: Poloxamer; PEI: Polyethyleneimine; BSA: Bovine serum albumin; PEG: Polyethylene glycol; GET: Glycosaminoglycan (GAG)-binding enhanced transduction; DSPE_PEG: 1,2-Distearoyl-sn-Glycero-3-Phosphoethanolamine conjugated with Polyethylene glycol; pDNA: Plasmid DNA; CF: Cystic fibrosis.

**Table 4 pharmaceutics-16-00347-t004:** Relevant patents in the field of inhalable NCs, identified through a search conducted on the Espace Patent search website (https://worldwide.espacenet.com, accessed on 23 January 2024).

Publication No.	Date of Filing	Applicant	Title of the Invention	Formulation	Hypothesis/Application/Advantages
WO 2012/017406 A1	04/08/2011	Indian Institute of Technology (IIT), Bombay, Maharashtra, India	Exogenous pulmonary surfactant preparation comprising a phospholipid and an adjuvant.	Preparation of exogenous pulmonary surfactant and its use in the production of surface-active drug delivery systems (liposomes).	Relieved symptoms of breathlessness and corrected surfactant dysfunction in pulmonary tuberculosis at the pulmonary air–aqueous interface.
WO 2015/061467	22/10/2014	Shire Human Genetic Therapies Inc., Lexington, MA, USA; Massachusetts Institute of Technology, Cambridge, MA, USA.	Lipid formulations for the delivery of messenger RNA.	Cationic lipid-based liposomes encapsulating mRNA.	Lung administration of cationic lipid-based liposomes provided the expression of the protein encoded by the encapsulated mRNA in vivo.
EP 2 893 922 A1	09/01/2015	Heart Biotech Pharma Limited, London, UK.	Pharmaceutical formulations for the treatment of pulmonary arterial hypertension.	Polymeric nanoparticles, encapsulating a therapeutic agent suitable for the treatment of pulmonary arterial hypertension, embedded within crosslinked polymeric hydrogel microparticles.	Enhanced bioavailability, increased deep-lung targeting, reduced dose frequency, avoided macrophage clearance, and sustained pulmonary delivery of therapeutic agent suitable for the treatment of pulmonary arterial hypertension (prostacyclin analogues, nitric oxide, or PPAR β agonists) compared to free drugs.
WO 2017/098474 A1	09/12/2016	Universidade do Minho, Braga, Portugal; Universidade do Porto, Porto, Portugal; Instituto de Biologia Molecular E Celular IBMC, Porto, Portugal.	Antimicrobial peptide-loaded hyaluronic acid-based formulations, method of production and uses thereof.	Hyaluronic acid-based nanoparticles for the pulmonary delivery of antimicrobial peptides.	Safe and efficient delivery of antimicrobial peptides and low molecular weight pharmaceuticals to infected tissues, in particular lungs and airways, such as tuberculosis.
WO 2019/180047	19/03/2019	Algipharma AS, Sandvika, Norway.	Use of alginate oligomers to enhance the translocationof micro/nanoparticles across the mucus layers.	Cationic micro/nanoparticles modified or associated with alginate oligomer having at least 70% of mannuronate residues.	Reduction of cationic particulate systems’ interactions with lung fluids (i.e., mucus) and toxicity.
WO 2019/209787 A1	23/04/2019	TLC Biopharmaceuticals INC, San Francisco, CA, USA; Taiwan Liposome CO., Ltd, Taipei City, Taiwan.	Inhalable liposomal sustained release composition for use in treating pulmonary diseases.	Liposomes prepared using a PEG-modified lipid and encapsulating a tyrosine kinase inhibitor.	Aerosolizable liposomes able to provide consistent drug pharmacokinetic and pharmacodynamic profiles while achieving the desired efficacy and safety.
WO 2020/081974 A1	18/10/2019	Ohio State Innovation Foundation, Columbus, OH, USA.	NCs for lung inflammation therapy.	Extracellular vesicles, functionalized with lung-targeted ligands, containing anti-inflammatory cargo.	Targeted delivery of anti-inflammatory compounds.
WO 2020/212545 A1	17/04/2020	Fundacion CIDETEC, Donostia-San Sebastian, Gipuzkoa, Spain.	Nano-antibiotics based on single-chain dextran nanoparticles.	Nanoconjugates based on single-chain dextran methacrylate or acrylate for the encapsulation and delivery of hydrophilic antimicrobials.	Increased hydrophilic antibiotic loading, optimized drug release, and suitable distribution at the target site.
WO 2022/079105 A1	13/10/2021	Ludwig-Maximilians Universitat Munchen, Munchen, Germany; University of Columbia, New York, NY, USA.	Nano-in-Micro-encapsulated siRNA dry powders, method for producing the same, and use of a powder formulation.	Dry powder Polyelectolyte complexes, formed from at least a polyamine and/or a polyamide and/or polyester and siRNA.	A new method to produce nano-in-micro dry powder encapsulating siRNA, able to provide high quantity and integrity of the encapsulated molecule and high redispersibility of the nanoparticles.

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
