# Peer review of "State-of-the-Art Review on Inhalable Lipid and Polymer Nanocarriers: Design and Development Perspectives"

_pharmaceutics, 2024, doi:10.3390/pharmaceutics16030347_

Round 1

Reviewer 1 Report

Comments and Suggestions for Authors

     The manuscript aims to focus on the design and development of inhalable nanocarriers for respiratory diseases. It highlights the importance of customized drug delivery systems for inhalation therapy and provides an overview of advancements in nanocarrier development for delivering drugs to lung tissue, including overcoming lung barriers. The manuscript could provide valuable information that can contribute to this specific research area. However, the manuscript requires substantial improvement. Therefore, the following questions and suggestions should be addressed. Overall, I recommend that the manuscript undergo extensive revisions before publication.

  1. Please carefully make the reference citation in a consistent format. Some citation numbers currently are placed before the period (dot) symbol, while others are placed after the period symbol.
  2. In Line 35, please correct the number 1 formatting.
  3. References must be included in the text. Specifically, Lines 40-42 and 113-135. Figures that are duplicated, modified, or cropped from other sources should be acknowledged.
  4. Please make the following correction in Table 2 (Reference 181): after the semicolon, please change the capital "C" to a lowercase "c" in "Cyto-". Remove the underline located after the period (".") in Line 805.
  5. The manuscript only mentions two types of nanocarriers: lipid-based and polymer-based nanocarriers. What about inorganic nanomaterials? What factors are being considered, specifically excluding other nanomaterials? These factors can be addressed in the content.
  6. The manuscript title includes the term "respiratory diseases". The manuscript discusses nanocarriers and biological barriers. However, it is unclear whether respiratory diseases can be associated with nanocarriers and barriers. Therefore, it is important to define which specific respiratory diseases will be discussed in the manuscript.
Comments on the Quality of English Language

Minor editing of English language required.

Author Response

Referee 1:

The manuscript aims to focus on the design and development of inhalable nanocarriers for respiratory diseases. It highlights the importance of customized drug delivery systems for inhalation therapy and provides an overview of advancements in nanocarrier development for delivering drugs to lung tissue, including overcoming lung barriers. The manuscript could provide valuable information that can contribute to this specific research area. However, the manuscript requires substantial improvement. Therefore, the following questions and suggestions should be addressed. Overall, I recommend that the manuscript undergo extensive revisions before publication.

1)Please carefully make the reference citation in a consistent format. Some citation numbers currently are placed before the period (dot) symbol, while others are placed after the period symbol.

We appreciate the reviewer for the comment. All citations have been positioned before the period (dot) symbol.

2)In Line 35, please correct the number 1 formatting.

We have likewise corrected the formatting.

3)References must be included in the text. Specifically, Lines 40-42 and 113-135.

In response to the referee's recommendation, we have incorporated the omitted references into the respective sections.

4)Figures that are duplicated, modified, or cropped from other sources should be acknowledged.

We thank the reviewer for this comment. We have modified the figure 1 and all figures included in the new version of the manuscript are now original and not duplicated, modified, or cropped from other sources.

5)Please make the following correction in Table 2 (Reference 181): after the semicolon, please change the capital "C" to a lowercase "c" in "Cyto-".

The correction has been implemented.

4)Remove the underline located after the period (".") in Line 805.

The correction has been implemented.

5)The manuscript only mentions two types of nanocarriers: lipid-based and polymer-based nanocarriers. What about inorganic nanomaterials? What factors are being considered, specifically excluding other nanomaterials? These factors can be addressed in the content.

The referee is right. We have added a short section (Lines 66-71) regarding the use of inorganic nanomaterials for inhalation therapy, including recent references that can provide guidance in case the readers are interested in the topic. However, considering the significant concerns of toxicity that limit the application of inorganic particles in pulmonary delivery in the clinic, we did not go further in depth. The manuscript focuses on lipid- and polymer-based nanocarriers. This has been better highlighted also in the Review’s Title.

6)The manuscript title includes the term "respiratory diseases". The manuscript discusses nanocarriers and biological barriers. However, it is unclear whether respiratory diseases can be associated with nanocarriers and barriers. Therefore, it is important to define which specific respiratory diseases will be discussed in the manuscript.

We appreciate the reviewer's suggestion, although we respectfully disagree with the proposal to define a more specific target respiratory disease. Our primary focus is on nanocarriers and their current state of advancement. As a result, we have opted to remove 'respiratory diseases' from the title to ensure a more accurate alignment between the outline and the title.

Reviewer 2 Report

Comments and Suggestions for Authors

Dear Authors,

A state-of-the-art review on the design and development of inhalable nanocarriers for respiratory diseases is an interesting and exciting article.

1. The introduction needs to be rewritten carefully to cover all aspects of the title and outline. More references need to be added to cover physiological and formulation aspects.

2. The Patents section needs to be added to the manuscript.

3. The conclusion should be the answer to a hypothesis and novelty.

4. The figure needs to be added to each nanocarrier section.

5. Challenges for respiratory diseases need to be addressed carefully.

Comments on the Quality of English Language

NA

Author Response

Referee 2:

Dear Authors,

A state-of-the-art review on the design and development of inhalable nanocarriers for respiratory diseases is an interesting and exciting article.

7) The introduction needs to be rewritten carefully to cover all aspects of the title and outline.

We appreciate the reviewer's suggestion and we have carefully implemented the introduction section according to our primary focus, which is on nanocarriers and their current state of advancement. In order to increase the adherence between the title and the outline, we have opted to change it as well.

8) More references need to be added to cover physiological and formulation aspects.

We would like to thank the referee for the insightful comment. We have included new references pertaining to physiological and formulation aspects in the new version of the manuscript.

9) The Patents section needs to be added to the manuscript.

We appreciate the reviewer for the valuable suggestion, and we have added a new section (paragraph 7, in the new version of the manuscript) discussing recent patents on 'inhalable nanocarriers'. Additionally, a table summarizing key information from each reported patent has been included.

10) The conclusion should be the answer to a hypothesis and novelty.

The conclusion section was implemented according to reviewer’s suggestions.

11) The figure needs to be added to each nanocarrier section.

According to the reviewer’s suggestion, a new figure representing a schematic diagram on different architectures of polymer-based nanocarriers was added (Figure 3 in the new version of the manuscript).

12) Challenges for respiratory diseases need to be addressed carefully.

We appreciate the reviewer's suggestion, although we respectfully disagree considering that our main emphasis is on nanocarriers rather than a particular respiratory disease. Therefore, we have opted to remove "respiratory disease" from the title to better align with our primary focus.

Reviewer 3 Report

Comments and Suggestions for Authors

The submitted manuscript is a comprehensive review that describes the current state of the art in the field of inhaled delivery systems.

Readers will be very interested in the manuscript. The advantages include a vast number of relevant references, high-quality analysis of published works, and nice graphic design.

Before the article is accepted for publication, the authors would like to include a part in which they would examine the prospects for the development of inhaled delivery systems.

It would also be useful to include a section analyzing patent activity in this field.

Author Response

Referee 3:

The submitted manuscript is a comprehensive review that describes the current state of the art in the field of inhaled delivery systems.

Readers will be very interested in the manuscript. The advantages include a vast number of relevant references, high-quality analysis of published works, and nice graphic design.

13) Before the article is accepted for publication, the authors would like to include a part in which they would examine the prospects for the development of inhaled delivery systems.

We appreciate the reviewer’s valuable suggestion. Considering the enhanced focus on future perspectives in the development and characterization of inhalable nanoparticulate systems within the conclusions section, we found it fitting to consolidate conclusions and perspectives into a single, comprehensive paragraph (paragraph 8, revised manuscript).

14)It would also be useful to include a section analyzing patent activity in this field.

We thank the reviewer for the suggestion, and we have added a new section (paragraph 7, in the new version of the manuscript) discussing recent patents on 'inhalable nanocarriers'. Additionally, a table summarizing key information from each reported patent has been included.

Round 2

Reviewer 1 Report

Comments and Suggestions for Authors

     The manuscript has undergone significant improvements after the revision. Overall, only minor revisions are necessary before publishing. The following questions and suggestions should be addressed.

1. In Line 49-51, add a reference.

2. In Line 68, "Excellent" is not the best choice of word. It would be preferable to use "detailed", "exhaustive", "full", or "thorough". It’s a different situation than the same word, Excellent, used in Line 105. Please review the entire manuscript carefully.

3. In Line 81, use the word 'larger' instead of the symbol (>) in the sentence.

4. In Line 87 and 88, please use the consistent dash symbol for size range.

5. In line 123, add references following "Figure 1". Other works have previously addressed the lung barriers. This also applies to Figures 2 (Line 347) and 3 (Line 550).

Comments on the Quality of English Language

Minor editing of English language required

Author Response

Point-by point Response to Reviewer’s comments

We thank very much the referee for his/her valuable comments, which assisted the authors in improving the paper. As explained in the following, all the Referees’ suggestions were taken into consideration in revising the manuscript.

Referee 1:

The manuscript has undergone significant improvements after the revision. Overall, only minor revisions are necessary before publishing. The following questions and suggestions should be addressed.

1.In Line 49-51, add a reference.

The correction has been implemented.

  1. In Line 68, "Excellent" is not the best choice of word. It would be preferable to use "detailed", "exhaustive", "full", or "thorough". It’s a different situation than the same word, Excellent, used in Line 105. Please review the entire manuscript carefully.

We appreciate the reviewer for the comment and we changed the word choice both for line 68 and 105.

  1. In Line 81, use the word 'larger' instead of the symbol (>) in the sentence.

The correction has been implemented.

  1. In Line 87 and 88, please use the consistent dash symbol for size range.

The correction has been implemented.

  1. In line 123, add references following "Figure 1". Other works have previously addressed the lung barriers. This also applies to Figures 2 (Line 347) and 3 (Line 550).

The correction has been implemented.
